# The Transmembrane Protease Serine 2 (TMPRSS2) Non-Protease Domains Regulating Severe Acute Respiratory Syndrome Coronavirus 2 (SARS-CoV-2) Spike-Mediated Virus Entry

**DOI:** 10.3390/v15102124

**Published:** 2023-10-19

**Authors:** Romano Strobelt, Julia Adler, Yosef Shaul

**Affiliations:** Department of Molecular Genetics, Weizmann Institute of Science, Rehovot 7610001, Israel

**Keywords:** SARS-CoV-2 infection, SARS-CoV-2 entry, TMPRSS2 viral entry, TMPRSS2-Spike interaction, mechanism viral entry, receptor-mediated endocytosis, endosomal infection

## Abstract

The severe acute respiratory syndrome coronavirus 2 (SARS-CoV-2) enters cells by binding to the angiotensin-converting enzyme 2 (hACE2) receptor. This process is aided by the transmembrane protease serine 2 (TMPRSS2), which enhances entry efficiency and infectiousness by cleaving the SARS-CoV-2 surface glycoprotein (Spike). The cleavage primes the Spike protein, promoting membrane fusion instead of receptor-mediated endocytosis. Despite the pivotal role played by TMPRSS2, our understanding of its non-protease distinct domains remains limited. In this report, we present evidence indicating the potential phosphorylation of a minimum of six tyrosine residues within the cytosolic tail (CT) of TMPRSS2. Via the use of TMPRSS2 CT phospho-mimetic mutants, we observed a reduction in TMPRSS2 protease activity, accompanied by a decrease in SARS-CoV-2 pseudovirus transduction, which was found to occur mainly via the endosomal pathway. We expanded our investigation beyond TMPRSS2 CT and discovered the involvement of other non-protease domains in regulating infection. Our co-immunoprecipitation experiments demonstrated a strong interaction between TMPRSS2 and Spike. We revealed a 21 amino acid long TMPRSS2-Spike-binding region (TSBR) within the TMPRSS2 scavenger receptor cysteine-rich (SRCR) domain that contributes to this interaction. Our study sheds light on novel functionalities associated with TMPRSS2’s cytosolic tail and SRCR region. Both of these regions have the capability to regulate SARS-CoV-2 entry pathways. These findings contribute to a deeper understanding of the complex interplay between viral entry and host factors, opening new avenues for potential therapeutic interventions.

## 1. Introduction

The severe acute respiratory syndrome coronavirus 2 (SARS-CoV-2) is the causative agent behind the outbreak of the coronavirus disease 19 (COVID-19). Since its initial emergence in Wuhan in December 2019, millions have succumbed to the effects of COVID-19 [1,2,3,4]. SARS-CoV-2 represents the third coronavirus outbreak characterized by a high mortality rate, following the occurrences of SARS-CoV-1 and the Middle East respiratory syndrome coronavirus (MERS-CoV). This situation raises alarms about the potential for future coronavirus pandemics [5,6,7].

SARS-CoV-2 gains entry into cells through its surface glycoprotein, Spike, by attaching to its receptor, angiotensin-converting enzyme 2 (hACE2). To facilitate the fusion of the virus membrane with the host cell or endosome membrane, Spike must undergo cleavage at two distinct sites: the polybasic cleavage site (S1/S2) and the transmembrane serine protease 2 (TMPRSS2) cleavage site (S2’). This cleavage process leads to the release of the Spike fusogenic peptide [5,8,9,10]. The hACE2 receptor alone is sufficient to enable SARS-CoV-2 infection. In the absence of TMPRSS2, SARS-CoV-2 virions enter the cells through the endosomal pathway, where Spike is cleaved by endosomal cathepsin-L. However, in the presence of TMPRSS2, the membrane fusion entry pathway becomes predominant due to Spike cleavage at the cell surface. This entry process is faster than the endosomal pathway, resulting in increased infection kinetics and higher viral load [9,11,12,13]. Furthermore, TMPRSS2 facilitates syncytia formation, thereby enhancing virion-free cell-to-cell spread [14,15].

Interestingly, in TMPRSS2-positive cells, the compound camostat, which inhibits TMPRSS2 enzymatic activity, effectively inhibits SARS-CoV-2 infection via both the membrane and the endosomal cell entry routes. Therefore, it has been proposed that both entry pathways, namely TMPRSS2-mediated membrane fusion and TMPRSS2-independent receptor-mediated endocytosis, are mutually exclusive. However, the underlying mechanism remains enigmatic [9,11,16]. Notably, since TMPRSS2 is highly expressed in lung cells, this may explain why the exclusive endosomal entry inhibitor hydroxychloroquine (HCQ) failed in treating COVID-19 patients [17,18,19,20,21].

In addition to the critical role of TMPRSS2 protease activity, we hypothesized that other domains of this protein may play roles in regulating the viral entry pathway into cells. Here, we report that the cytosolic tail (CT) of TMPRSS2 can undergo posttranslational tyrosine phosphorylation. Intriguingly, when we engineered a phosphomimetic TMPRSS2-CT mutant, we observed a significant reduction in TMPRSS2 enzymatic activity. This reduction in enzymatic activity, in turn, resulted in a marked decrease in SARS-CoV-2 pseudovirus transduction, specifically via the endosomal pathway. Consequently, we proposed the hypothesis that TMPRSS2 physically interacts with Spike in a manner that limits its accessibility to the endosomal pathway. We found a 21-amino acid (aa) region within the scavenger receptor cysteine-rich domain (SRCR) of TMPRSS2 that binds to Spike. Our analysis revealed that TMPRSS2-Spike physical interaction facilitates membrane infection, provided that TMPRSS2 is enzymatically competent.

## 2. Materials and Methods

### 2.1. Cell Culture

Human embryonic kidney cells expressing SV40 large T-antigen (HEK293T, ATCC^®^, American Type Culture collection, Manassas, VA, USA) were cultured in Dulbecco’s modified eagle’s medium (Gibco^®^, Thermo Scientific^®^, Waltham, MA, USA) supplemented with 100 units/mL penicillin and 100 µg/mL streptomycin (pen/strep; Biological Industries^®^, Beit Haemek, Israel), and 8% fetal bovine serum (Gibco^®^). Via transduction of the hACE2-4xMyc gene in HEK293T and selection with 15 µg/mL blasticidin, we received HEK293T-hACE2 as previously described [22]. To create HEK293T-hACE2 with the constructed TMPRSS2-mutants, HEK293T-hACE2 were transfected with the plasmids 48 h before the experimental setting. Cells were non-enzymatically harvested with phosphate-buffered saline (PBS) containing 1 mM EGTA before every experiment. Between passages, cells were harvested with Trypsin B solution (Biological Industries^®^).

### 2.2. Plasmids and Cloning

The plasmid pCG1-SARS-CoV-2-Spike-HA was generously provided by Stefan Pöhlmann, and the pCMV3-SARS-CoV-2-Spike plasmid was kindly provided by Ron Diskin; both contain the Spike Wuhan-Hu-1 aa sequence. In the case of various TMPRSS2 constructs, the pEFIRES-TMPRSS2-Flag plasmid served as the foundational backbone. PCR products were designed to overlap with either the beginning or the end of the respective cloning/mutation site. These PCR products were then employed as templates for a second round of PCR to generate a complete TMPRSS2-Flag product, complete with the desired mutations. The NheI and XbaI restriction sites were utilized for the cloning of the TMPRSS2 construct back into the pEFIRES plasmid. From Anthony Koleske, we received pDsRed-Abl2 plasmid, and we replaced the C-terminal RFP reporter with a stop-codon via the NotI and AgeI restriction sites. The procedures for cloning pBiFC-Jun-YFPn, pBiFC-Fos-YFPc, pCDNA-Δ81-Abl1, pCDNA3.1-Delta-Spike, and pCDNA3.1-Omicron-Spike, and respective aa 655 mutants were previously documented [22,23,24]. All used oligos can be found in Appendix A.

### 2.3. Transfection and Pseudovirus Transduction

The transfection and transduction methods were previously described [22]. In brief, the calcium phosphate (CaPO_4_) method was employed for all transfections. For 6 cm plates, the prepared DNA mix consisted of 8 µg of DNA and 25 µL of 2.5 M calcium chloride (CaCl_2_) in a total of 250 µL of water. In the subsequent step, the DNA mix was vortexed, and 250 µL of HEPES-buffered saline (HBS2x) was added dropwise. After a one-minute incubation, the transfection mix was introduced to cells that were 80% confluent. The medium was replaced with fresh growth medium after eight hours. For the production of lenti-pseudoviruses, HEK293T was transfected with pCMV-Δ19Spike (1.5 µg), pGIPZ-tGFP (3.5 µg), and pCMV-ΔR8.9 (3 µg) plasmids in a 6 cm plate, as previously described [22]. The lenti-virus-containing medium was filtered through a 0.45 µm membrane filter (Sartorius^®^, Beit Haemek, Israel) after 2.5 days. Cells successfully transduced with Lenti-Spike contained a tGFP-reporter and were treated with 5 µg/mL Hoechst solution (Molecular Probes^®^, Life Technologies^®^, Carlsbad, CA, USA) before being analyzed under a microscope. The ratio of Hoechst-stained nuclei (representing all cells) to GFP-emitting cells (indicating infected cells) was quantified using a previously described ImageJ macro [22].

### 2.4. Immunoprecipitation (IP) and Immunoblotting (IB)

Cells were harvested using ice-cold PBS, and then centrifuged at 1500× *g* for 5 min. The resulting pellet was resuspended in RIPA buffer containing both protease inhibitors (ApexBio^®^, Houston, TX, USA) and tyrosine-phosphatase inhibitors (Sigma^®^, Merck^®^, Rehovot, Israel) at a 1:100 concentration ratio each. After a 15 min incubation on ice, the cell lysate underwent a 15 min centrifugation at maximum speed. The supernatant was subsequently combined with a three-times concentrated Laemmli buffer at the appropriate concentration.

For IP, the supernatant was mixed with either Flag- or HA-antibody-conjugated beads (Sigma^®^) and incubated for 4 h at 4 °C on a rotator. Subsequent washing steps and elution (Sigma^®^) were carried out following the manufacturer’s protocol to separate HA- or Flag-tagged proteins and their respective interacting partners from other proteins. After an additional brief centrifugation at 13,000 rpm, the IP samples were mixed with Laemmli buffer.

The IB samples were boiled for two minutes prior to being loaded onto an 8% gel, if not described differently. Subsequently, standard procedures for SDS-PAGE, blotting, and antibody treatment were followed as previously reported [25]. For enhancing the signal of horseradish peroxidase-conjugated secondary antibodies (Jackson Immuno Research Laboratories^®^, West Grove, PA, USA) and visualization, an EZ-ECL kit (Biological Industries^®^) was used. IB-signal was recorded using ImageQuant LAS 4000 (GE Healthcare, Piscataway, NJ, USA). We used primary antibodies against c-abl-K12, phosphorylated-tyrosine/PY20 (Santa Cruz^®^, Santa Cruz, CA, USA), Abl2 (Biolegend^®^, London, UK), tubulin (Sigma^®^), actin, HA, Flag (Sigma^®^), and Myc (9E10; Weizmann Institute, Rehovot, Israel).

### 2.5. Serine Protease Enzymatic Assay

The TMPRSS2-IP lysate was combined with PBS supplemented with a fluorogenic peptide substrate, BOC-QAR-AMC (100 mM, R&D Systems^®^, Abingdon, UK), in a total volume of 50 µL within the wells of a 96-well plate. Fluorescence was continuously monitored for three hours at five-minute intervals using an excitation wavelength of 365 nm and an emission wavelength of 410 nm.

### 2.6. Structure Modeling

All structural models were visualized using PyMol 2.5.5 software. The structure of TMPRSS2 protein was downloaded from alphafold.ebi.ac.uk and represented a computational prediction model. The Spike-Omicron structure was shared by the National Center for Biotechnology Information (NCBI) under the PDB-ID: 7TGW and is based on a cryo-EM structure from Ye et al. [26].

### 2.7. Graphs and Statistics

GraphPad Prism 9.1 software was utilized for generating all graphs and conducting statistical analyses. Error bars in the graphs represent the standard error of the mean (SEM). Unless otherwise stated, all experiments report the mean of three independent biological experiments. To perform statistical tests, the standard deviation of the reference bar (control bar) was calculated from the three technical replicates of each biological experiment. All Student’s *t*-tests were two-tailed.

## 3. Results

### 3.1. Tyrosine Phosphorylation of TMPRSS2 Cytosolic Tail Reduces Its Protease Activity

Compared to other virus entry-associated serine proteases, such as TMPRSS4 and TMPRSS11a (https://www.uniprot.org, accessed on 24 September 2023), TMPRSS2’s cytosolic tail (CT) is distinguished by its abundance of tyrosine residues [27,28,29,30]. According to data from Phosphosite.org, specific tyrosine residues within TMPRSS2 CT, namely Y44, Y45, and Y52, were identified as phosphorylation sites. Notably, this region contains an Abelson kinase (Abl)-specific Y45xxP motif, as well as two PxxP motifs. The Y45xxP motif is well-suited for possible Abl1/2 phosphorylation, while the presence of the two PxxP motifs suggests their potential to interact with Abl1/2 (Figure 1a) [31]. To explore this possibility, we transfected HEK293T with either constitutively active Δ81Abl1 [32] or Abl2 along with TMPRSS2-Flag. The deletion of the first 81 aa is necessary for Abl1 to prevent the self-inhibition of its enzymatic activity [32]. Subsequent immunoprecipitation (IP) of TMPRSS2-Flag demonstrated the formation of a complex between Abl1 and TMPRSS2 (Figure 1b). Immunoblotting (IB) of IP samples with PY20, a phosphorylated tyrosine-specific antibody, revealed TMPRSS2 tyrosine phosphorylation. As positive controls, we used the yes-associated protein (Yap), a reported Abl1 substrate [33], and cortactin, a reported Abl2-substrate [34,35]. TMPRSS2 was tyrosine phosphorylated by Abl1 but poorly, if any, by Abl2.

To pinpoint the tyrosine residues that undergo phosphorylation, we employed mutagenesis by substituting tyrosine with phenylalanine (Y→F). Phenylalanine shares a similar chemical structure with tyrosine but lacks the ability to be phosphorylated. We observed a significant reduction in the tyrosine phosphorylation of TMPRSS2, specifically when six residues were Y→F mutated (Figure 1a,c). As expected, phosphorylation was inhibited by imatinib, an Abl kinase inhibitor. The TMPRSS2 CT region is highly disordered, rendering it accessible to the relevant tyrosine kinases (Figure 1d). These findings suggest that TMPRSS2 is susceptible to tyrosine phosphorylation and point towards Abl1 as a potential tyrosine kinase involved in this process.

To evaluate the impact of TMPRSS2 CT tyrosine phosphorylation on TMPRSS2 protease enzymatic activity, phospho-mimetic (Y→D) or phospho-dead (Y→F) TMPRSS2 mutants were overexpressed in HEK293T cells and purified via Flag-IP. Next, the purified protein samples were mixed with the serine protease-sensitive BOC-QAR-AMC, a synthetic serine protease substrate emitting a fluorescence signal upon its cleavage, and measured fluorescence intensity (Figure 1e) [36,37]. All TMPRSS2 Y→F and the 2xD mutants were enzymatically active (Figure 1f). However, only poor enzymatic activity was observed with the TMPRSS2-6xD mutant. These data suggest that tyrosine phosphorylation of TMPRSS2 CT modulates TMPRSS2 protease enzymatic activity.

**Figure 1 viruses-15-02124-f001:**
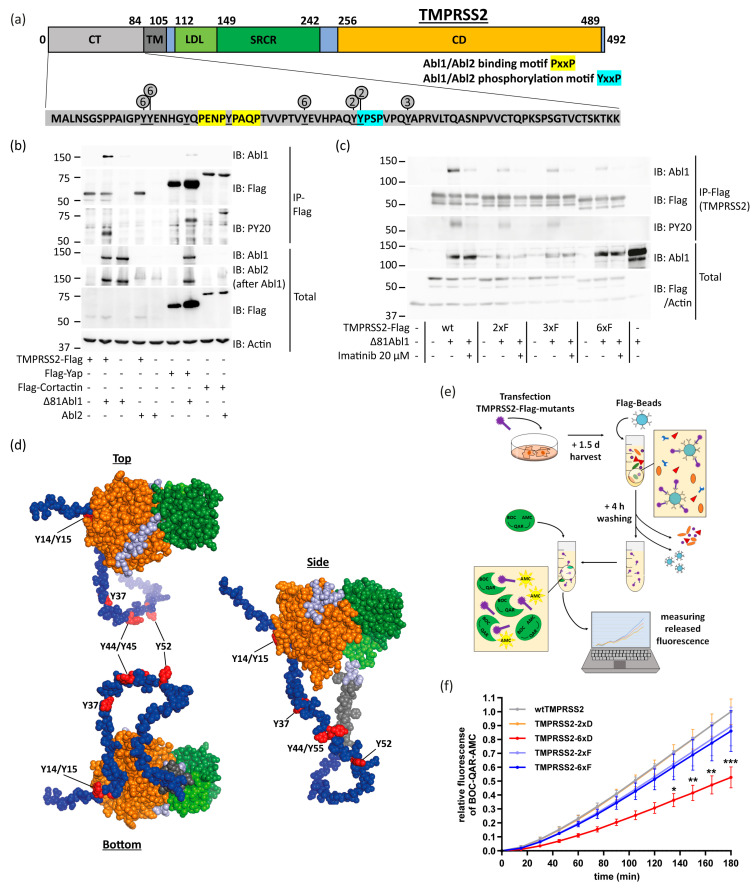
Abl1 can phosphorylate up to six tyrosine residues within the TMPRSS2 cytosolic tail and decrease its enzymatic activity. (**a**) The membrane protein TMPRSS2 consists of 492 amino acids (aa) and has several functional regions. The N-terminus 1–84 region is the cytosolic tail (CT), followed by the transmembrane domain (TM). The 112–149 region is the LDL-receptor class A domain (LDL), and the 149–242 region is the scavenger receptor cysteine-rich domain (SRCR). The catalytic domain (CD), 256–489 region, is the serine protease domain of the S1 family [38]. TMPRSS2 CT contains eight tyrosine residues (Y; underlined) and two potential Abl1/Abl2 binding motifs PxxP (yellow). Consensus Abl1/Abl2 phosphorylation motif YxxP is highlighted by cyan. Y residues that were either Y→F or Y→D mutated are shown by circles. The numbers in the circles classified the 2x, 3x, and 6x Y→F or Y→D mutants. (**b**) TMPRSS2 is phosphorylated by Abl1 but not Abl2. HEK293T were transfected with the respective plasmids, and extracts of the transfected cells were immunoprecipitated with anti-Flag beads and immunoblotted (IB) with the respective antibodies 2.5 d later. Yap and cortactin served as positive controls for Abl1 and Abl2 substrates, respectively. PY20 is a phosphorylated tyrosine-specific antibody. The result was confirmed in an additional experiment. (**c**) Abl1 phosphorylates up to six TMPRSS2 CT tyrosine residues. HEK293T were transfected with the indicated TMPRSS2 mutants shown in panel a and, after 24 h, were treated with DMSO or 20 µM imatinib. The next day, cells were harvested and analyzed as above. The result was confirmed in two additional experiments. (**d**) TMPRSS2 structure was taken from a prediction model of alphafold.ebi.ac.uk (accessed 10 September 2023) and modified with PyMol. The disordered CT region is highlighted in blue, the TM in dark gray, the LDL in bright green, the CT in dark green, and the CD in orange. The discussed tyrosine residues are highlighted in red. (**e**) Scheme of serine protease assay. Cells were transfected with TMPRSS2-Flag mutants, harvested after 1.5 d, and TMPRSS2-Flag mutants were purified using Flag-beads. The purified TMPRSS2-mutants were mixed with serine protease-sensitive fluorogenic peptide substrate, BOC-QAR-AMC. The released fluorogenic signal was measured every 15 min and represents levels of enzymatic activity. (**f**) TMPRSS2 mutant 6xD has low enzymatic activity. HEK293T were transfected with indicated TMPRSS2-Flag mutants and treated as described above. The increase in fluorescence rate was determined for each mutant and subsequently converted to the ratio of wtTMPRSS2 at 180 min. The graph represents three experiments and was statistically analyzed by two-way-ANOVA with multi-comparison and Tukey’s post-test. * *p* ≤ 0.05; ** *p* ≤ 0.01; *** *p* ≤ 0.001.

### 3.2. TMPRSS2-6xD Mutant Poorly Supports Spike-Mediated Infection and Membrane Fusion Entry

Next, we investigated the effect of TMPRSS2-6xD on pseudo-SARS-CoV-2 transduction. We transduced HEK293T-hACE2 expressing the corresponding TMPRSS2 mutants with SARS-CoV-2 pseudovirus [22] and evaluated the transduction rate after 2 days. The transduction efficacy was lower in TMPRSS2-6xD than that of TMPRSS2-negative cells (Figure 2a), attributing a suppressive role to this mutant. These results suggest that TMPRSS2 phospho-mimetic mutant suppresses SARS-CoV-2 pseudovirus transduction.

Virions enter TMPRSS2-positive cells through the membrane fusion pathway, while TMPRSS2-negative cells primarily utilize receptor-mediated endocytosis as the dominant entry route. These distinct entry mechanisms can be effectively inhibited using the serine protease inhibitor camostat for membrane fusion and the cathepsin-L inhibitor E64d for endocytosis [9,22]. Given the reduced enzymatic activity of TMPRSS2-6xD, the efficacy of camostat in preventing lenti-Spike transduction is expected to be minor. Indeed, pretreatment with camostat led to a decrease in lenti-Spike transduction in wild-type TMPRSS2 cells, while it did not affect infectivity in cells lacking TMPRSS2 or expressing TMPRSS2-6xD (Figure 2b). Interestingly, TMPRSS2-6xD-mediated low level of transduction was markedly sensitive to the endosomal cathepsin-L inhibitor E64d [39], suggesting that lenti-Spike enters TMPRSS2-6xD cells via endocytosis instead of membrane fusion (Figure 2c). Under this condition, a residual level of infection was maintained that might result from the involvement of some other proteases or even TMPRSS2-6xD itself.

In SARS-CoV-2-infected cells, cytopathic syncytia formation occurs due to the membrane fusion of cells expressing Spike and cells expressing hACE2, a process that is further accelerated by the co-expression of TMPRSS2 [11,15,22]. Cell-cell fusion rate, as measured using the bimolecular fluorescence complementation (BiFC) approach [22], was lower in TMPRSS2-6xD compared with wtTMPRSS2 cells (Figure 2d). However, it was significantly higher compared to TMPRSS2-negative cells, possibly because TMPRSS2-6xD maintained a low level of enzymatic activity. These results suggest that TMPRSS2-6xD poorly supports SARS-CoV-2 membrane fusion and membrane fusion entry.

### 3.3. TMPRSS2 Physically Interacts with Spike

In search of additional TMPRSS2 regulatory regions, we asked whether TMPRSS2 physically interacts with Spike. To this end, we constructed several TMPRSS2 C-terminal truncation mutants for co-immunoprecipitation experiments (Figure 3a). Interestingly, a robust binding between TMPRSS2 and Spike was revealed (Figure 3b). The TMPRSS2 amino acid 1–405 region exhibited comparable efficacy to the wild type in its ability to immunoprecipitate Spike. In contrast, regions 1–316 and 1–170 demonstrated reduced effectiveness but still managed to pull down certain levels of Spike. However, no Spike IP was evident by the TMPRSS2 1–159 region. Similar results were obtained in a reciprocal experiment using Spike to bring down TMPRSS2 and the corresponding truncated mutants (Appendix A). These results suggest that TMPRSS2 binds Spike, and an extended TMPRSS2 region is required for achieving maximal binding.

Inspection of the TMPRSS2 protein via uniport.org revealed its cysteine-rich nature supports multiple disulfide bridges interlinking nearly all regions of the protein, except for a 21 aa region within the identified area, which we designate as TMPRSS2-Spike-binding region (TSBR) (Figure 3c). Next, the 21 aa long TSBR was deleted to construct TMPRSS2-Δ149-170. Surprisingly, unlike the wt, the TMPRSS2-Δ149-170 mutant was inactive in cleaving Spike to form the TMPRSS2-specific S2′ Spike fragment (Figure 3d). Notably, the expression level of the TMPRSS2-Δ149-170 mutant was much higher than that of wt TMPRSS2 and accompanied by a slight increase in Spike expression. Despite the high level of TMPRSS2-Δ149-170, the level of co-immunoprecipitation was comparable to that of wt TMPRSS2 (Figure 3d). These data suggest that TMPRSS2-Δ149-170 mutant displays a diminished binding to Spike, highlighting an inefficiency in their interaction. Structural prediction analysis has unveiled that the region spanning amino acids 149 to 170 is prominently exposed on the protein surface, facilitating interaction with potential binding partners (Figure 3e).

Next, we assessed the TMPRSS2-Δ149-170 on supporting transduction. The transduction of Spike-lenti pseudovirus was significantly increased in the presence of TMPRSS2 (Figure 3f). In contrast, TMPRSS2-Δ149-170 did not increase transduction efficiency. These data suggest that TSBR deleted TMPRSS2 poorly supports transduction. To investigate the route of infection of the TMPRSS2-Δ149-170 mutant, cells were E64d treated, the inhibitor of the endosomal pathway [39]. While the transduction of the wt TMPRSS2 cells was E64d refractory, a marked reduction was observed in cells expressing TMPRSS2-Δ149-170 (Figure 3f). These results suggest that TSBR mediating Spike interaction regulates TMPRSS2 enzymatic activity and membrane route of infection.

### 3.4. TMPRSS2-6xD Physically Interacts with Spike

We next asked whether the TMPRSS2-6xD suppressive effect on transduction could derive from abortive Spike interaction. We addressed this possibility by investigating Spike-TMPRSS2 binding. HEK293T was transfected with Spike and TMPRSS2 constructs and harvested for TMPRSS2-IP and IB 1.5 days later. Neither the TMPRSS2-specific cleavage pattern of Spike nor the amount of TMPRSS2-Spike-associated proteins were substantially changed (Figure 4a). Remarkably, both Spike-TMPRSS2 and TMPRSS2-6xD robustly bind Spike even after Spike cleavage, as evident from the appearance of Spike S1/S2 and S2′ fragments in TMPRSS2-IP lysate.

To confirm that TMPRSS2-6xD is properly localized in the cells and extracellularly exposed, we conducted experiments to show whether it binds Spike at the cell surface like wtTMPRSS2. To this end, we performed attachment assays where HEK293T cells overexpressing Spike were mixed with HEK293T-hACE2 overexpressing TMPRSS2 in a ratio of 1:1. After a short incubation time, 1 h, extracts were prepared and subjected to HA-Spike-IP (Figure 4b). The incubation for this short period assures investigation of the attachment process, without further processing, in line with cell-cell fusion assays described above (Figure 2d). Both wt TMPRSS2 and TMPRSS2-6xD were comparably immunoprecipitated (Figure 4c). Moreover, the levels of expression of TMPRSS2-6xD and wtTMPRSS2 were found to be similar and consistent with the IB data presented above (Figure 4a). These results suggest TMPRSS2-6xD and wtTMPRSS2 are similar at the levels of cleavage pattern, expression, Spike interaction, and cellular localization.

### 3.5. Omicron Spike Y655 Residue Reduces TMPRSS2 Binding

Omicron exhibits reduced TMPRSS2-dependency, enabling infection of TMPRSS2-positive cells through both entry pathways, facilitated by the presence of Omicron-specific Spike Y655 residue (Figure 5a). However, the precise underlining mechanism remains elusive [13,24,40]. To investigate whether the diminished Omicrons reduced TMPRSS2-dependency might stem from a decreased binding between Omicron-Spike and TMPRSS2 due to the Y655 residue, HEK293T were transfected with Omicron-Spike wt and Y655H mutant, along with a reciprocal experiment involving Delta-Spike wt and H655Y. The aim was to assess the impact of these mutations on TMPRSS2 binding. The analysis of HA-TMPRSS2 immunoprecipitation revealed the levels of the Omicron Y655H mutant were elevated, while the Delta H655Y mutant exhibited reduced levels relative to its wt counterpart (Figure 5b). These results strongly indicate the participation of the wt Spike residue H655 in binding to TMPRSS2. Consequently, these findings point towards a significant role of the interaction between TMPRSS2 and the H655 residue of the Spike protein in defining TMPRSS2-dependency. Furthermore, they suggest that a robust binding between Spike and TMPRSS2 counteracts the endosomal entry of the SARS-CoV-2 virus (Figure 5c).

## 4. Discussion

Here, we elucidate the role of TMPRSS2 non-protease domains in regulating SARS-CoV-2 lenti-Spike infection. While the catalytic domain of TMPRSS2 is well documented for its pivotal role in mediating the SARS-CoV-2 membrane route of infection, the significance of other regions within TMPRSS2 has received limited attention [9,41,42,43]. Given that TMPRSS2 plays a crucial role in facilitating the cellular entry of various viruses, including but not limited to influenza, parainfluenza, hepatitis C, and multiple coronaviruses, conducting an examination of TMPRSS2’s distinct domains bears significant relevance in understanding and combating these viral infections [42,44,45,46].

We identified TMPRSS2 cytosolic tail (CT) as a putative substrate for phosphorylation by tyrosine kinases. Using a phosphomimetic mutation approach, our study demonstrates that tyrosine phosphorylation of the TMPRSS2 CT can downregulate its enzymatic activity. This allosteric regulation of TMPRSS2 activity via modification of its cytosolic tail might have some implications in the development of new antiviral strategies. While camostat and nafamostat are general serine protease inhibitors [9,16], targeting TMPRSS2 phosphorylation may offer a more virus-specific approach. This is particularly relevant since other virus-associated proteases, such as TMPRSS4 and TMPRSS11a, have limited phosphorylatable amino acid residues in their cytosolic tails [29,30]. Furthermore, it is worth noting that TMPRSS2-ERG fusion transcripts are frequently described in prostate cancer. In these cases, the N-terminal portion of TMPRSS2, including the cytosolic tail but not the catalytic domain, is commonly fused to the ERG transcription factor in cancer patients [47,48,49]. This underscores the importance of a comprehensive understanding of the interplay between tyrosine kinases and TMPRSS2, which could pave the way for innovative treatment strategies targeting both viral infections and cancer.

The authenticity of endogenous Abl1 as the genuine TMPRSS2 kinase is currently uncertain, given that our study relied on the over-expression of an overactive Δ81Abl1 mutant. The definitive identification of the genuine tyrosine kinase responsible for TMPRSS2 phosphorylation remains a topic for future investigations. In our prior study [22], we documented the reduction in SARS-CoV-2 infection with the use of imatinib in an Abl1/Abl2-independent process. In this study, we demonstrate that imatinib effectively inhibits Abl1-mediated hyperphosphorylation of TMPRSS2, suggesting an enhancement in SARS-CoV-2 infection control. However, it is worth noting that imatinib’s inhibitory effect primarily occurs at the level of Spike protein by direct imatinib-Spike interaction [22], which is an upstream process. This overrides the significance of TMPRSS2 phosphorylation in the context of SARS-CoV-2 infection.

SRCR domains are known to interact with extracellular proteins and molecules, but this domain is not well characterized in the context of TMPRSS2 [38,50]. Interestingly, TMPRSS2 gene single-nucleotide polymorphism (SNP) rs12329760 causing V160M mutation in TMPRSS2-SRCR domain is associated with severe COVID-19 disease progression, suggesting that the SRCR domain impacts SARS-CoV-2 pathogenicity [51,52]. We describe here that TMPRSS2 robustly binds Spike. Truncated deletion mutant analysis revealed that the aa 149–170 region within the TMPRSS2-SRCR domain plays a role in Spike binding. However, given that the TMPRSS2-Δ149-170 deletion mutant continues to bind Spike, we hypothesize that the TMPRSS2 binding region may be more extensive, possibly encompassing an additional binding region. This putative second binding region is likely situated between TMPRSS2 amino acids 316 and 405, as its deletion resulted in reduced binding. It would be intriguing to investigate whether TMPRSS2 can bind to the surface proteins of other TMPRSS2-dependent viruses, potentially establishing a broader role for the Spike-binding domain in viral infection.

A mediator of TMPRSS2 interaction with SARS-CoV-2 Spike protein is the H655 residue. The observation that Omicron infection is only partially dependent on TMPRSS2 might be explained by the weakened TMPRSS2 interaction due to the Omicron Spike Y655 residue [13,24,53]. Considering the observed correlation between TMPRSS2 affinity for Spike and the infection pathway via the cell membrane, we hypothesize that the physical interaction between these two proteins reduces the utilization of the endosomal route for SARS-CoV-2 Spike-mediated entry (Figure 6).

## Figures and Tables

**Figure 2 viruses-15-02124-f002:**
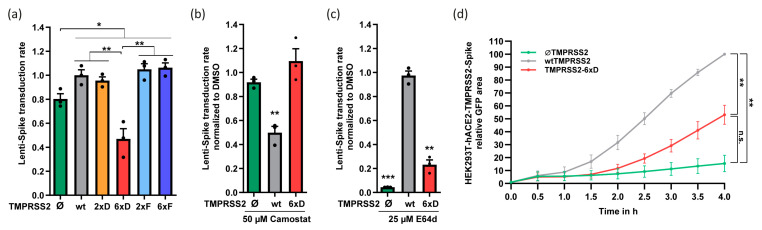
The TMPRSS2-6xD phospho-mimetic mutant reduces SARS-CoV-2 pseudovirus transduction and does not support membrane-fusion entry. (**a**) The presence of TMPRSS2 increases infection of lenti-Spike, but it is compromised in TMPRSS2-6xD cells. HEK293T-hACE2 were transfected with indicated TMPRSS2 mutants and selected with 1.2 µM puromycin after 1.5 d. The next day, cells were infected with SARS-CoV-2 pseudovirus, and the medium was changed to fresh growth medium after 8 h. After 1.5 d, cells were treated with Hoechst and transduction efficiency was calculated by the ratio between all cells (blue) and infected cells (green) by microscope and ImageJ analysis [22]. The transduction efficiency was normalized to cells with wtTMPRSS2. (**b**) Lenti-Spike transduction in HEK293T-hACE2-TMPRSS2-6xD is not inhibited by camostat. HEK293T-hACE2 were transfected with respective TMPRSS2 construct and selected with 1.2 µM puromycin. Cells were treated with DMSO or 50 µM camostat for two h and were then infected with lenti-Spike. Further treatment and calculation of the transduction rate followed the same protocol as above. (**c**) Lenti-Spike transduction in HEK293T-hACE2-TMPRSS2-6xD follows the endocytosis pathway. The experiments were constructed as the previous experiment, but cells were treated either with DMSO or 25 µM E64d 2 h before lenti-Spike transduction. (**d**) TMPRSS2-6xD led to reduced fusion efficiency compared to wtTMPRSS2. HEK293T-hACE2 cells were transfected with wtTMPRSS2 or TMPRSS2-6xD and Jun-YFP_n_, while HEK293T cells were transfected with Spike and Fos-YFP_c_. After 1.5 d, cells were mixed in a 1:1 ratio, and images were taken with the IncuCyte system at half-hour intervals. IncuCyte Software (version 2020A) determined the total YFP area (µm^2^/Image), and the obtained data were normalized to 3 h of wtTMPRSS2 cells. The area under the curve was calculated for statistical evaluation, and the graph summarizes three biological replicates. For statistical evaluation, we used Student’s *t*-test. * *p* ≤ 0.05; ** *p* ≤ 0.01; *** *p* ≤ 0.001.

**Figure 3 viruses-15-02124-f003:**
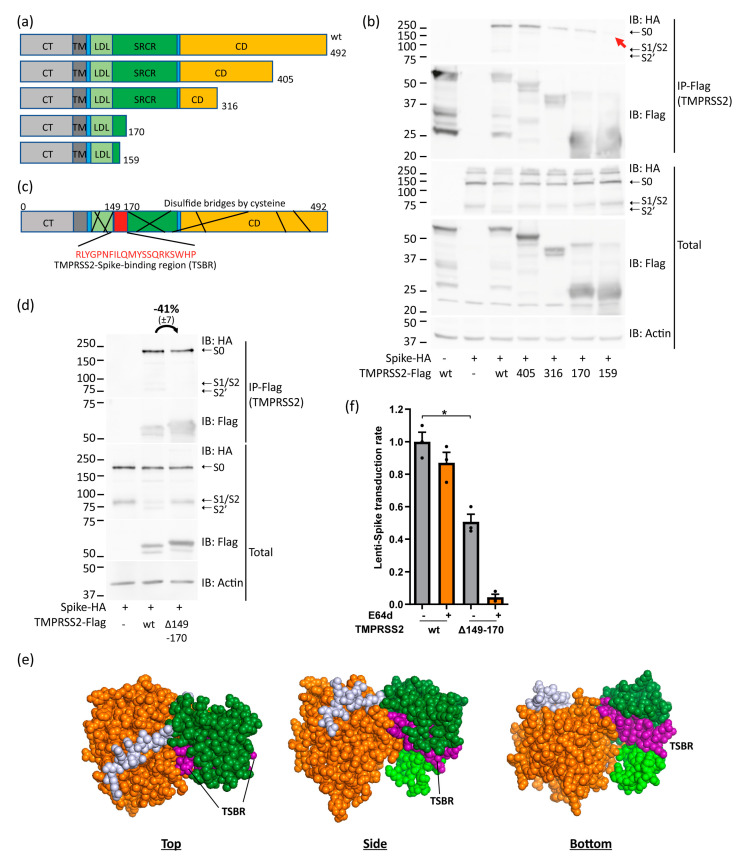
TMPRSS2 physically interacts with Spike. (**a**) Scheme of the constructed TMPRSS2 C-terminal truncation constructs. (**b**) Mapping the Spike-TMPRSS2 interacting region. HEK293T was transfected with the indicated plasmids and immunoprecipitated with Flag-beads, SDS-page on a 12% gel, and IB was performed as above. The red arrow shows the expected location of the Spike-HA band within the IP-Flag blot. (**c**) Scheme of TMPRSS2 protein and reported disulfide bridges by cysteine. Uniprot.org describes 9 different disulfide bonds within TMPRSS2 protein represented by black lines within the scheme. The 21 aa long TSBR lacks interconnection by cysteine bridges. (**d**) TSBR deleted TMPRSS2 is active in Spike-binding. HEK293T was transfected with the indicated constructs. HA-beads were used for IP of the Spike protein, followed by SDS-PAGE and WB. Two additional experiments verified the results. The measurements represent the ratio of band intensity between wt and Δ149-170 TMPRSS2. Band intensities of the three experiments were measured via ImageJ (v.1.48) and adjusted to the respective actin band, normalized to wtTMPRSS2, and compared to the IP/Total ratio. (**e**) TSBR is located on the cell surface. TMPRSS2 structure was taken from an alphafold.ebi.ac.uk (accessed on 10 September 2023) prediction and modified with PyMol. For better visualization, the TMPRSS2 CT and TM were removed. While the color code represents the same pattern as above, the TSBR within TMPRSS2 SRCR is highlighted in purple. (**f**) Cells expressing the TMPRSS2-Δ149-170 mutant did not increase SARS-CoV-2 pseudovirus transduction. HEK293T-hACE2 were transfected with the indicated TMPRSS2 plasmids, selected 1.5 d later overnight, and E64d-treated 2 h before and during transduction. The infectivity was normalized to wt TMPRSS2 cells. Student-*t*-test was used for statistical evaluation. * *p* ≤ 0.05.

**Figure 4 viruses-15-02124-f004:**
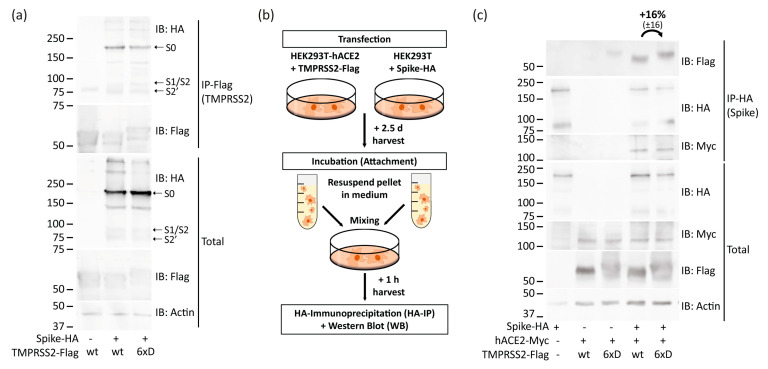
TMPRSS2-6xD mutant binds Spike. (**a**) HEK293T were transfected with Spike-HA together with either wtTMPRSS2-Flag or TMPRSS2-6xD-Flag. About 1.5 d later, cells were harvested, immunoprecipitated using anti-Flag, and afterward subjected to SDS-PAGE and IB. Highlighted band S0 indicates full-length Spike, while S1/S2 indicates the truncated C-terminal Spike peptide cleaved by furin or cathepsin-L and S2′ by TMPRSS2. The result was confirmed in two additional experiments. (**b**) Scheme of attachment assay. HEK293T-hACE2 was transfected with TMPRSS2-Flag, and HEK293T was transfected with Spike-HA and harvested after 2.5 d. Cell suspensions were centrifuged down, and the pellets were resuspended in a growth medium before mixing in a 1:1 ratio. After an hour, cells were harvested with RIPA buffer, and their extracts were subjected to IP and IB. (**c**) Both wt TMPRSS2 and TMPRSS2-6xD are located at the outer cell membrane and attached to Spike. HEK293T-hACE2 and HEK293T were treated as described above. The result was verified with an additional experiment, and measurements of band intensities were conducted like above, representing the IP/Total ratio normalized to wt TMPRSS2.

**Figure 5 viruses-15-02124-f005:**
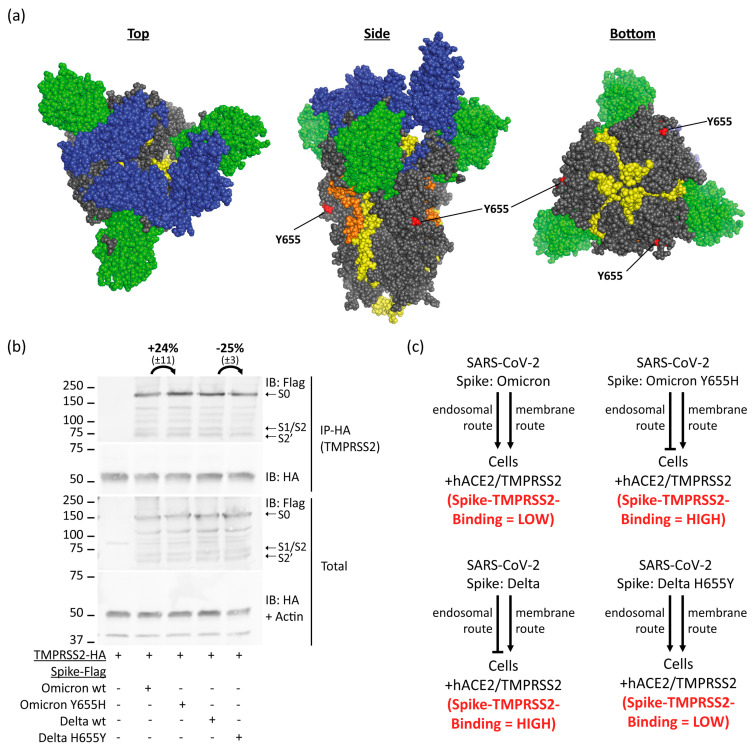
Spike residue Y655 determines affinity to TMPRSS2. (**a**) The Omicron Spike Y655 is highlighted on the Spike structure in red. Spike Omicron trimer structure was predicted by Ye et al. and was edited by us with PyMol [26]. Spike N-terminal domain is highlighted in green, the receptor-binding domain in blue, the fusion peptide in orange, and both heptad repeats in yellow. (**b**) Spike residue Y655 determines affinity to TMPRSS2. HEK293T-hACE were transfected with TMPRSS2 and respective Spike constructs. HA-IP to pulldown TMPRSS2 was performed, and samples were treated as described above. Omicron-Spike Y655H mutation increases TMPRSS2 binding while Delta-Spike H655Y mutation decreases TMPRSS2 binding compared to respective wt. Band intensities of the three replicates from panel 3c were measured via ImageJ and adjusted to the respective actin band. The Omicron samples were normalized to band intensity of wt Delta-Spike and Delta samples to wt Omicron-Spike. Afterward, the ratio of IP/Total was compared between respective wt Spike and belonging mutant. (**c**) Scheme of SARS-CoV-2 routes of infection in different conditions. SARS-CoV-2 enters hACE2-positive cells through receptor-mediated endocytosis and shifts entirely to membrane-fusion route when TMPRSS2 is additionally present. Omicron and respective subvariants are the only variants known to infect TMPRSS2-positive cells via both entry routes due to Omicron-Spike-specific Y655 residue. The extent of Spike-TMPRSS2 interaction determines the route of infection.

**Figure 6 viruses-15-02124-f006:**
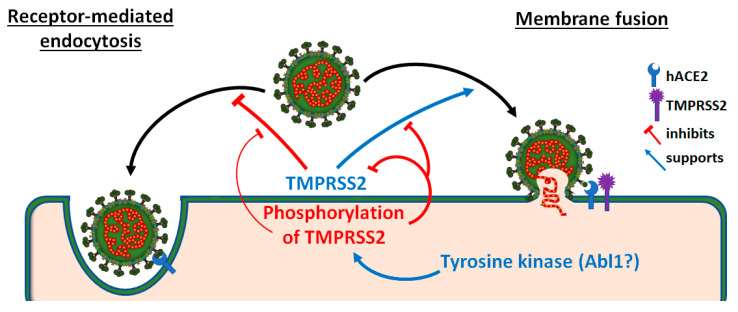
A summary model. SARS-CoV-2 binding to hACE2 enables two distinct entry routes: receptor-mediated endocytosis and membrane fusion. The presence of TMPRSS2 tilts the preference towards the membrane fusion pathway, likely facilitated by the physical binding of Spike. In contrast, the phosphorylation of TMPRSS2’s CT region by tyrosine kinases, such as Abl1, reduces TMPRSS2’s enzymatic activity. This, in turn, inhibits the membrane fusion pathway, leading to a decrease in the rate of SARS-CoV-2 infection.

## Data Availability

Not applicable.

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
