# Peer review of "The Transmembrane Protease Serine 2 (TMPRSS2) Non-Protease Domains Regulating Severe Acute Respiratory Syndrome Coronavirus 2 (SARS-CoV-2) Spike-Mediated Virus Entry"

_viruses, 2023, doi:10.3390/v15102124_

Round 1

Reviewer 1 Report

Strobelt and colleagues investigated the interactions between TMPRSS2 and SARS-CoV-2 S and their effect on S-mediated entry which can either occur via fusion at the cell membrane or via endocytosis followed by fusion with intracellular membranes. The authors showed that phosphorylation of the cytoplasmic tail of TMPRSS2 reduces enzymatic activity and thereby decreases S-mediated entry via membrane fusion. By the generation and overexpression of truncated versions of TMPRSS2, they further identified two regions within the protease that interact with SARS-CoV-2 S.

As a major point, the material and methods section lacks important information on reagents and protocols:

1.       There is no information on the generation of pseudoviruses. Based on the methods, it is unclear which kind of pseudoviruses has been used, which reporters are expressed and how transduction efficiency was evaluated.

2.       The list of plasmids is incomplete, e.g. Abl2 and del81Abl1 are missing.

3.       Which Spike protein has been used? Information on isolate is missing.

4.       Line 114: Which antibodies?

Additional points:

5.       It seems that the protein band in Fig. 1c are shifted to the right. First line (IB:Abl1): there is a strong protein band for Abl1+Imatinib, a faint band for no Abl1 and no band for Abl1 only. Second line (IB:Flag): There is a shift in molecular weight from TMPRRS only to TMPRRS+Abl1. Please check if especially line 1 and 2 are in the correct position according to the labelling.

6.       Did you test the 3xF/D mutants of TMPRSS2 regarding their enzymatic activity and effect on S-mediated transduction efficancy?

7.       The authors switch between “infection” and “transduction” when referring to pseudoviruses. For replication deficient pseudoviruses, the term “transduction” should be used.

8.       Fig. 3d), IB:Flag total: Why does the deletion mutant has a higher molecular weight?

9.       Line 330-332: The cellular localization of TMPRSS2 wt and mutant should be determined by microscopy.

Author Response

Thank you for your revision. Please see the attached Word file.

Reviewer 2 Report

In the present manuscript, Strobelt et al studied the non-protease distinct domains of TMPRSS2. They found there was a “six tyrosine residues” within the CT of TMPRSS2. They saw a reduction of TMPRSS2 protease activity together with a decrease in SARS-CoV-2 pseudovirus infection. They also found a 21 AA long TMPRSS2-Spike-binding region (TSBR) within SRCR domain, which contributes to the interaction between Spike and TMPRSS2.

Major Comments:

1.   The authors addressed that the phosphorylation of TMPRSS2's CT region reduces its enzymatic activity thus inhibiting the membrane fusion pathway, leading to a decrease of SARS-CoV-2 infection efficiency. However, there was no direct evidence to show decrease of SARS-CoV-2 infection efficiency was resulted by blockage of membrane fusion pathway. Are there any assays to evaluate the changed viral entry pathways directly?

2. The authors could not make a conclusion in Figure 1 that Abl1 was responsible for the phosphorylation of TMPRSS2. The data of interaction between Abl1 and TMPRSS2 was not enough to support this conclusion. Can the authors knockdown Abl1 using a different cell line and detect the changes of TMPRSS2 phosphorylation?

Minor comments:

1. It will be better if the authors change title as “The TMPRSS2 Non-Protease Domains Regulates SARS-CoV-2 Spike Mediated Virus Entry”

2. Lines 125-126, please check the font settings “c-abl-K12, phosphorylated-tyrosine/PY20 (Santa Cruz®), Abl2 (Bio- 125legend®), tubulin (Sigma®), actin, HA, Flag (Sigma®) and Myc”

3. Line 157, please give a brief introduction of Δ81Abl1.

4. Figure 1B. Immunoblotting of IP samples with Abl2 antibody was missing.

5. Figure 1C. Please double check if the orders of each lane match each other. It seems wt Flag-TMPRSS2 could not pull down Abl1 which is opposite as the findings in figure 1b. Also the data showed that there was enhanced interaction between Abl1 and TMPRSS2 when phosphorylation of TMPRSS2 was inhibited by imatinib.

Author Response

(The authors gave the same response as above.)

Round 2

Reviewer 1 Report

The authors have addressed my previous points. I have no comments to add.

Reviewer 2 Report

The authors responded all my comments. I don't have any other comments.